# Connection between the Gut Microbiota of Largemouth Bass (*Micropterus salmoides*) and Microbiota of the Pond Culture Environment

**DOI:** 10.3390/microorganisms9081770

**Published:** 2021-08-19

**Authors:** Qianfu Liu, Zini Lai, Yuan Gao, Chao Wang, Yanyi Zeng, Erchun Liu, Yongzhan Mai, Wanling Yang, Haiyan Li

**Affiliations:** 1Pearl River Fisheries Research Institute, Chinese Academy of Fishery Sciences, Guangzhou 510380, China; liuqfwh@163.com (Q.L.); gaoyuan0328@163.com (Y.G.); chaowang80@163.com (C.W.); yanyizeng84@163.com (Y.Z.); liuerchun2021@163.com (E.L.); yongzhanmai@prfri.ac.cn (Y.M.); yangwanling@prfri.ac.cn (W.Y.); lihaiyan@prfri.ac.cn (H.L.); 2Guangzhou Scientific Observing and Experimental Station of National Fisheries Resources and Environment, Guangzhou 510380, China; 3Fishery Ecological Environment Monitoring Center of Pearl River Basin, Ministry of Agriculture and Rural Affairs, Guangzhou 510380, China; 4Guangdong Provincial Key Laboratory of Aquatic Animal Immune Technology, Guangzhou 510380, China; 5Colllege of Fisheries and Life, Shanghai Ocean University, Shanghai 201306, China

**Keywords:** gut microbiota, aquaculture, environment microbiota, largemouth bass

## Abstract

The vital role of the gut microbiota in fish growth, development, immunity, and health has been largely confirmed. However, the interaction between environmental microbiota and the gut microbiota of aquaculture species remains unclear. Therefore, we analyzed the gut microbiota of largemouth bass (*Micropterus salmoides*) collected from subtropical ponds in southern China, as well as the pond water and aquatic sediment microbiota, using high-throughput sequencing of the 16S rRNA gene. Our results demonstrated significant differences in the compositions of pond water, sediment, and the gut microbiota of largemouth bass. Moreover, these compositions changed throughout the culture period. Only approximately 1% of the bacterial species in the pond sediment and gut microbiota were exchanged. However, the bacterial proportion of the gut microbiota from pond water microbiota was approximately 7% in samples collected in June and August, which increased markedly to 73% in October. Similarly, the proportion of bacteria in the pond water microbiota from the gut microbiota was approximately 12% in June and August, which increased to 45% in October. The study findings provide basic information for understanding the interactions between environmental microbiota and the gut microbiota of cultured fish, which may contribute to improved pond culture practices for largemouth bass.

## 1. Introduction

With capture fishery production being relatively static since the late 1980s, aquaculture has played a crucial role in maintaining sufficient fish supply to meet increasing consumer demand [1]. Global fish production was estimated to have reached approximately 179 million tons in 2018, of which aquaculture accounted for 46% and 52% of the total production and human consumption, respectively [2]. China has remained a major fish producer, accounting for 35% of the global fish production in 2018 [2]. Pond culture is the main form of aquaculture in China, accounting for 37.22% of the total aquaculture area and 48.84% of the total output of aquatic products in China [3].

The development of intensive aquaculture technology has considerably improved the output of aquatic products per unit area of pond. However, the rapid growth of high-density, high-yield pond aquaculture has increased the pollution of aquatic environments [4]. Approximately 75% of feed nitrogen and phosphorus are not utilized, thereby persisting in the aquatic environments as waste [5,6]. These nutrients subsequently change the composition of the water microbiota, which may affect the gut microbiota of aquaculture species.

The gut microbiota play an important role in various physiological processes, thereby affecting the growth, development, immunity, and health of the host [7,8,9]. Therefore, the composition of the fish gut microbiota and their influencing factors have been widely investigated [10,11,12,13,14,15,16]. The composition and influencing factors of environmental microbiota in aquaculture have also been studied [17,18]. However, the interactions between environmental microbiota and the intestinal microbiota of cultured fish have not received equal attention, especially in aquaculture.

Therefore, to clarify these interactions, we analyzed the gut microbiota of largemouth bass (*Micropterus salmoides*) from subtropical ponds in southern China as well as the pond water and aquatic sediment microbiota, using high-throughput sequencing of the 16S rRNA gene. This study helps to elucidate the relationship between environmental microbiota and the gut microbiota of cultured fish, and the findings would eventually contribute to the development of effective pond management strategies and to the establishment of a healthy culture of largemouth bass.

## 2. Materials and Methods

### 2.1. Sample Collection

Water, sediment, and largemouth bass samples were collected from three outdoor aquaculture ponds (113°9′40″ E, 22°49′18″ N) located in Xingtan, China on 3 June (early culture period), 28 August (middle culture period), and 22 October (post-culture period). The three pond areas were approximately 3300, 4000, and 5200 m^2^ and the annual production was approximately 20,000, 20,000, and 27,000 kg, respectively. The depth of the pond water was approximately 2.5 m. The ponds were managed daily, and natural lighting conditions were adopted. During the experimental period, the fish were fed commercial puffed compound feed for largemouth bass (Guangdong Haida Group Co., Ltd., Guangzhou, China) at 9:00 a.m. and 4:00 p.m. at a daily rate of 3% of body weight. The water present in the ponds was not changed during the experiment. For each sampling, a 5 L plexiglass water sampler was used to collect water 0.5 m under the water surface. The water samples were then transported to the laboratory under transportation conditions at 4 °C for analysis. Sediment samples (approximately 5 g) were collected from three different locations in the three ponds, placed into 50 mL sterile centrifuge tubes, transported to the laboratory under transportation conditions at 4 °C, and stored at −80 °C for DNA extraction. Six largemouth bass were collected from each pond for body length and weight measurements, and three fish were randomly selected and transported to the laboratory under transportation conditions at 4 °C. The fish were anesthetized using an overdose of neutralized MS222 (ethyl 3-aminobenzoate methanesulfonic acid) and were subsequently dissected under sterile conditions. Hindgut samples (approximately 0.5 g) were collected in 2 mL sterile Eppendorf tubes and stored at −80 °C.

### 2.2. Determination of Aquatic Physicochemical Factors

Water transparency (SD) was measured in the field using a Secchi disk at each sampling site according to a standard method [19]. Water temperature (WT), pH, dissolved oxygen (DO), oxidation-reduction potential (ORP), conductivity (Cond), and total dissolved solids (TDS) were measured in the field using the ProQuatro smart portable multiparameter water quality analyzer (YSI, Yellow Springs, OH, USA). Concentrations of chlorophyll a (Chla), ammonium nitrogen (NH_4_^+^-N), nitrate nitrogen (NO_3_^−^-N), nitrite nitrogen (NO_2_^−^-N), total nitrogen (TN), and total phosphorus (TP) were determined according to a previously described method [20]. The permanganate index (COD_Mn_) was determined according to the standard method (ISO 8467:1993). The concentration of unionized ammonia (NH_3_) was calculated based on pH, WT, and the concentration of NH_4_^+^-N according to a previously described method [21].

### 2.3. Microbial DNA Extraction and High-Throughput Sequencing

Before the performance of DNA extraction, 500 mL of each water sample was filtered using a GF/C membrane with a 0.22 μm pore size (Whatman, Maidstone, UK) to collect and isolate the microorganisms present in the water, after which the filter membrane was cut and used to conduct microbial DNA extraction using the DNeasy PowerSoil kit (Qiagen, Hilden, Germany) according to the manufacturer’s instructions. The same method was used to conduct microbial DNA extraction from fish intestines and from approximately 0.5 g of the sediment sample.

The V4–V5 hypervariable region of the prokaryotic 16S rDNA was amplified using primers 515F and 909R with a 12 nt sample-specific barcode sequence included at the 5′-end of the 515F primer, as per methods previously described [22]. The amplicons were purified, quantified, and sequenced using the HiSeq 3000 platform (Illumina, San Diego, CA, USA) at Guangdong Meilikang Bio-Science, Ltd. (Dongguan, China), as per methods previously described [22,23]. Raw reads were merged using the Flash 1.2.8 software [24] and subsequently processed as per protocols previously described [25]. Briefly, all merged sequences were assigned to each sample based on their barcode sequences, and data on trimmed barcode and primer sequences were removed using the QIIME 1.9.0 software [26]. Chimeric sequences were identified, and data were removed using the Uchime algorithm [27]; thereafter, the high-quality sequences were clustered into operational taxonomic units (OTUs) at 97% identity using the UPARSE clustering algorithm [28]. Taxonomic assignment of each OTU was determined using the RDP classifier [29]. The alpha-diversity indices were calculated using the QIIME 1.9.0 software.

### 2.4. Data Analysis

Data are presented as mean ± standard error for each group. Non-parametric multivariate analysis of variance (PERMANOVA) [30] was used to determine significant differences in microbiota composition among groups using the vegan package in R [31]. Principal coordinate analysis (PCoA) of microbiota composition was conducted using the QIIME 1.9.0 software. The Kruskal–Wallis H-test was conducted using the STAMP software [32] to perform screening for significantly different taxa among the groups. Source tracking analysis was conducted using the SourceTracker package in R. A *p*-value < 0.05 was considered statistically significant.

## 3. Results

### 3.1. Changes in Physicochemical Factors of Aquaculture Pond Water, and Body Length and Weight of Largemouth Bass

The water temperature decreased significantly in October compared to the water temperatures in June and August, and the findings were consistent with the climatic conditions of the sampling area (Appendix A and Figure 1A). The pH, TDS, and DO levels fluctuated significantly during the experimental period (*p* < 0.05; Figure 1B–D). 

Although the TN content displayed a gradual upward trend, a significant difference was only determined between October and June (*p* < 0.05; Figure 1I). The NO_3_^−^-N content showed a decreasing trend during the experimental period (Figure 1J), while the concentrations of NO_2_^−^-N and NH_4_^+^-N increased gradually (Figure 1K,L). This might be attributable to the continuous accumulation of organic matter (Figure 1N) in the pond water during the 5-month aquaculture period, leading to increased oxygen consumption during decomposition of the organic matter, which resulted in a lack of oxygen availability at the bottom of the pond water and enhancement of the denitrification process. There were no significant changes in other physicochemical factors (Figure 1 and Appendix A), especially in the concentrations of TP and phosphate (Figure 1G,H).

The average body lengths and body weights of the largemouth bass were 10.70 ± 0.98, 20.76 ± 0.36, and 24.82 ± 0.72 cm, and 34.88 ± 10.10, 153.42 ± 11.72, and 265.21 ± 24.96 g in June, August, and October, respectively (Appendix A). These changes reflected growth of the cultured fish.

### 3.2. Differences between the Microbiota of Pond Water, Sediment, and the Largemouth Bass Gut

A total of 4,136,932 high-quality sequences were obtained for 81 samples (27 gut, 27 sediment, and 27 water samples). Finally, 23,430 high-quality sequences were randomly selected from each sample for subsequent analysis. Based on 97% sequence identity, these high-quality sequences were classified into 42,329 OTUs. The alpha-diversity index values (Kruskal–Wallis H-test, *p* < 0.05) and compositions (PERMANOVA, F = 23.19, *p* < 0.001; Figure 2A–G) differed significantly among the gut, pond water, and sediment microbiota. The number of OTUs and Shannon index values of the sediment microbiota were significantly higher than those of the water microbiota, followed by those of the gut microbiota (*p* < 0.05; Figure 1A,B). These results indicated that the coverage of the gut microbiota was the highest, while that of the sediment microbiota was the lowest (Figure 1F), and the finding was consistent with a previous report [33]. Within the same habitat (the gut, water, or sediment), the alpha-diversity index values of only a small number of samples differed significantly within the sampling times (Figure 2A–F). However, the microbiota composition of water (PERMANOVA, F = 1.72, *p* < 0.001), sediment (PERMANOVA, F = 2.15, *p* < 0.001), and the gut of largemouth bass (PERMANOVA, F = 1.92, *p* < 0.001) differed significantly among the different sampling times (Appendix A).

With the exception of a few OTUs, most OTUs were classified into 68 phyla (three archaeal and 65 bacterial phyla), among which Crenarchaeota, Acidobacteria, Actinobacteria, Bacteroidetes, Chlorobi, Chloroflexi, Cyanobacteria, Firmicutes, Fusobacteria, Gemmatimonadetes, NC10, Nitrospirae, OP8, Planctomycetes, Proteobacteria, Spirochaetes, Tenericutes, Verrucomicrobia, and WS3 dominated the microbiota (Figure 2H). These phyla comprised 98.00 ± 0.21% of the analyzed high-quality sequences. Significant differences in their relative abundances were observed among the different habitats (Figure 2H).

At the genus level, 1633 genera were detected, including 154 dominant genera. The most dominant genera differed significantly among the gut, sediment, and water microbiota (Figure 3A). Abundances of *Geobacillus*, *Lactobacillus*, *Clostridium*, *Cetobacterium*, *Burkholdena*, *Delftia*, *Citrobacter*, *Escherichia*, *Klebsiella*, *Plesiomonas*, *Mycoplasma*, and two unidentified genera from Chlamydomonadaceae and Streptophyta were significantly enhanced in the largemouth bass gut microbiota; abundances of *Gemmatimonas*, *Phenylobacterium*, *Rhodobacter*, *Novosphingobium*, *Giesbergeria*, *Hydrogenoptiaga*, *Limnohabitans*, *Rhodoterax*, *Variovorax*, and certain unidentified genera were significantly enhanced in the water microbiota; and abundances of *GOUTA19*, *Gallionella*, *Thiobacillus*, *Dok59*, *Sulfuritalea*, *Thauera*, *Geobacter*, and a few unidentified genera were significantly enhanced in the sediment microbiota (Figure 3A). In the gut microbiota, abundances of *Bacillus*, *Geobacillus*, *Staphylococcus*, *Lactococcus*, *Clostridium*, *Burkholderia*, *Escherichia*, *Kocuria*, and certain unidentified genera were significantly enhanced in the samples collected in June; abundances of *Synechococcus*, *Gallionella*, *Sulfuritalea*, *Geobacter*, *Plesiomonas*, *Psychrobacter*, *Flavobacterium*, and a few unidentified genera were significantly enhanced in the samples collected in August; and abundances of *Cetobacterium*, *Vibrio*, *Mycoplasma*, and an unidentified genus from Aeromonadaceae were significantly enhanced in the samples collected in October (Figure 3B). In the sediment microbiota, abundances of *Synechococcus*, *Gemmatimonas*, *Rhodobacter*, *Novosphingobium*, *Geobacter*, and certain unidentified genera were significantly enhanced in the samples collected in June; all significantly enhanced genera were unidentified genera in the samples collected in August; abundances of *Cetobactenum*, *Delftia*, *Diaphorobacter*, *Giesbergeria*, *Hydrogenophaga*, *Limnihabitans*, *Methylibium*, *Rhodoferax*, *Rubrivivax*, *Polynucleobacter*, *Vogesella*, *Dok59*, *Sulfurtalea*, *Thauera*, *Rheinheimera*, *Corynebacterium*, *Mycobacterium*, and certain unidentified genera were significantly enhanced in the samples collected in October (Figure 2C). In the water microbiota, abundances of *Synechococcus*, *Gemmatimonas*, *Ralstonia*, *Mycoplasma*, *Candidatus Aquiluna*, *and Candidatus Rhodoluna*, and certain unidentified genera were significantly enhanced in the samples collected in June; abundances of *Cetobacterium*, *GOUTA19*, *Plesiomonas*, and many unidentified genera were significantly enhanced in the samples collected in August; and abundances of *Planktothrix*, *Hydrogenophaga*, *Limnohabitans*, *Methylibium*, *Rhodoferax*, *Rubrivivax*, *Vogesella*, *Dok59*, *Rheinheimera*, *Pseudomonas*, *Flavobacterium*, and a few unidentified genera were significantly enhanced in the samples collected in October (Figure 3D).

### 3.3. Connection between the Gut Microbiota of Largemouth Bass and Pond Water and Sediment Microbiota

Source tracking analysis was conducted to analyze the relationship between the gut microbiota of largemouth bass and the pond water and sediment microbiota. The results based on all samples indicated that 7.03 ± 3.47% and 12.69 ± 3.63% of the bacterial composition in the gut microbiota could be attributable to the pond sediment and water microbiota, respectively. Additionally, 7.20 ± 2.85% of the bacterial composition in the sediment microbiota and 48.48 ± 2.27% of the bacterial composition in the water microbiota could be attributable to the gut microbiota, while approximately 30% of the bacterial composition in the pond water and sediment microbiota was subjected to exchange (Figure 4A). However, the results of each sampling revealed that only approximately 1% of the bacterial composition in the pond sediment and gut microbiota was subjected to exchange (Figure 4B–D,F). The proportion of bacteria in the gut microbiota from the pond water microbiota was approximately 7% in samples collected in June and August, while it increased markedly to 73.11 ± 4.08% in October (Figure 4B–E). The proportion of bacteria in the pond water microbiota that could be attributable to the gut microbiota was 11.82 ± 0.96% in June, 11.56 ± 1.68% in August, and the proportion increased to 44.79 ± 3.24% in October (Figure 4B–D). Except for the samples collected in August, the proportions of bacteria subjected to exchange between the pond water and sediment microbiota were similar throughout the experimental period, although the proportion subjected to exchange in October was significantly higher than that in June and August (*p* < 0.05; Figure 4B–D,G).

## 4. Discussion

The crucial role of the fish gut microbiome is widely recognized [9,34,35], which has led to the execution of studies investigating the composition of fish gut microbiota and its influencing factors [10,11,14,15,33,36,37,38]. However, except for a few reports describing the relationship between microbiota of the aquatic environment and the gut microbiota of aquatic animals [33,39], the interactions between environmental microbiota and the gut microbiota of aquaculture species remain understudied. Here, we clarified the relationship between the pond water and sediment microbiota and the gut microbiota of largemouth bass. Our results indicated that, in general, approximately 7.03 ± 3.47% of the bacterial composition in the gut microbiota of largemouth bass could be attributable to the sediment microbiota, while 12.69 ± 3.63% of the bacterial composition could be attributable to the pond water microbiota. However, according to sampling time analysis, only approximately 1% of the bacterial composition in the gut microbiota could be attributable to the sediment microbiota, and this proportion did not change during the cultivation period (Figure 4). Although the proportion of bacteria in the gut microbiota attributable to the pond water microbiota remained approximately 7% in June and August, it increased markedly to 73.11 ± 4.08% in October. Even though environmental factors and growth could cause the changes in the gut microbiota of fish [11,15], the reason for this change and its effect on the health and growth of largemouth bass warrants further study.

Habitat is the most important factor affecting the microbial community [33,39,40]. Previous studies have reported that the water microbiota of subtropical ponds in summer is dominated by Proteobacteria, Cyanobacteria, and Bacteroidetes [18], while the pond sediment microbiota is dominated by Proteobacteria, Chlorobi, TA06, and Fusobacteria, and the gut microbiota of fish is significantly enriched in Fusobacteria and Firmicutes [41]. Our results confirmed that the compositions of the water, sediment, and gut microbiota were significantly different (Figure 2G and Figure 3A), which corroborated previously reported results [41]. Apart from TA06 not being present as the dominant phylum and Fusobacteria not being significantly enriched in the sediment microbiota, the results of the current study were consistent with those of previous reports [18,41]. Moreover, the microbiota composition of these habitats changed during the culture period (Figure 3B–D and Appendix A). However, in general the gut microbiota composition changed significantly less than the microbiota composition of the pond water and sediment. These results suggest that although fish are poikilotherms and their gut microbiota are subjected to adaptations to changes in their external (such as water temperature) and internal (such as growth) environments, the gut microbiota composition of largemouth bass remains relatively stable, although the trend was not as remarkable as that for homoiothermic mammals. Proteobacteria, Fusobacteria, Firmicutes, and Cyanobacteria were the most prevalent phyla in the gut microbiota of largemouth bass, consistent with previous findings reported for other fish species [13,33,42,43].

Previous studies have indicated that lactic acid bacteria (*Lactobacillus*, *Streptococcus*, and *Lactococcus*) and *Bacillus* spp. are potential probiotic strains [33,44]. *Lactobacillus* abundance was enhanced in the gut microbiota of largemouth bass compared with that in the pond water and sediment. *Lactococcus* and *Bacillus* abundances were enhanced in the gut microbiota samples collected in June compared to those collected in August and October. On the other hand, pond water and sediment microbiota are generally considered an important source of pathogens in pond-cultured fish [33]. Indeed, the gut microbiota act as a reservoir for many opportunistic pathogens [33,45]. Among them, *Pseudomonas* and *Flavobacterium* are the two most important opportunistic fish pathogens, and they are highly abundant in the gut contents of grass carp [33]. Our results revealed that the abundances of these strains were significantly enhanced in the microbiota of water samples collected in October compared with those collected at other sampling times, while *Flavobacterium* abundance was significantly enhanced in the microbiota of gut samples collected in August compared with those collected at other sampling times (Figure 3). Although previous investigations have proposed that the origin and composition of the gut microbiota are attributable to their environment [33,46], our results suggested that under normal conditions, only a small proportion of bacteria from the pond water and sediment microbiota contributed to the composition and diversity of the fish gut microbiota (Figure 4). These results imply that microorganisms from the environment may constitute the fish gut microbiota establishment after hatching; however, the development of stable gut microbiota during fish growth limits further entry of microorganisms from the environment into the fish gut. The causes of this phenomenon warrant further study, along with ascertainment of the probability of an increase in the entry of pathogens present in pond water and sediment into the fish gut under abnormal conditions (such as lack of oxygen or high NH_4_^+^-N levels).

## 5. Conclusions

In conclusion, our results demonstrated that the microbiota compositions of the pond water, sediment, and the gut of largemouth bass differed significantly. Moreover, the microbiota compositions of these habitats changed during development of the pond culture. Proteobacteria, Fusobacteria, Firmicutes, and Cyanobacteria were the most prevalent phyla in the gut microbiota of largemouth bass. Only approximately 1% of the bacterial composition in the pond sediment and gut microbiota was subjected to exchange. However, the bacterial proportion of the gut microbiota attributable to pond water microbiota was approximately 7% in the samples collected in June and August, while it increased markedly to 73.11 ± 4.08% in October. Furthermore, the bacterial proportion of the pond water microbiota attributable to the gut microbiota was approximately 11.82 ± 0.96% in June, 11.56 ± 1.68% in August, and 44.79 ± 3.24% in October.

## Figures and Tables

**Figure 1 microorganisms-09-01770-f001:**
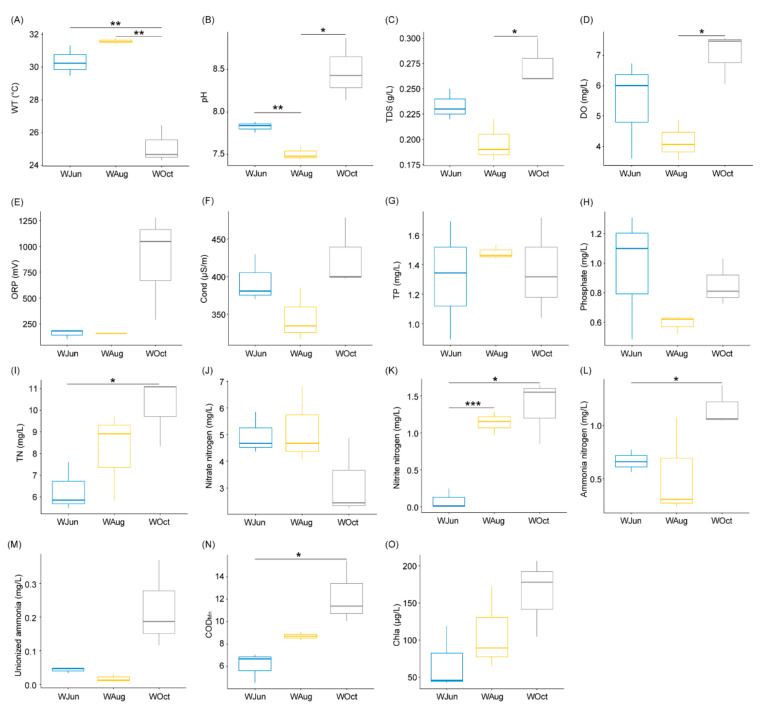
Water-associated physicochemical factors of aquaculture ponds. (**A**) WT, water temperature; (**B**) pH; (**C**) TDS, total dissolvable solids; (**D**) DO, dissolved oxygen; (**E**) ORP, oxidation reduction potential; (**F**) Cond, conductivity; (**G**) TP, total phosphorus; (**H**) phosphate; (**I**) TN, total nitrogen; (**J**) nitrate nitrogen; (**K**) nitrite nitrogen; (**L**) ammonia nitrogen; (**M**) unionized ammonia; (**N**) COD_Mn_, chemical oxygen demand; and (**O**) Chla, chlorophyll a. * *p* < 0.05; ** *p* < 0.01; *** *p* < 0.001.

**Figure 2 microorganisms-09-01770-f002:**
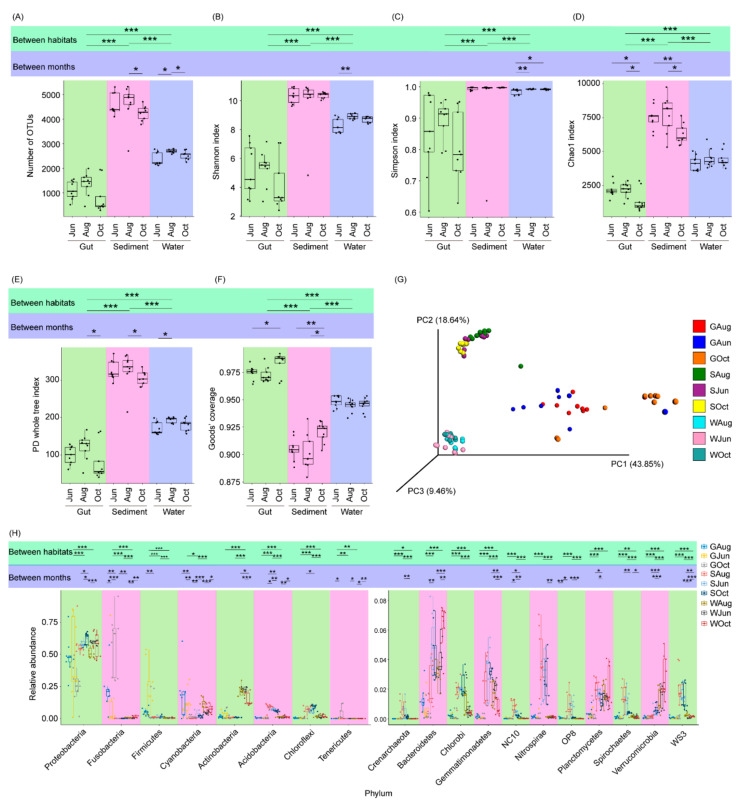
Differences between the microbiota of pond water, sediment, and the largemouth bass gut. (**A**) number of OTUs in the microbiota; (**B**) Shannon index of the microbiota; (**C**) Simpson index of the microbiota; (**D**) Chao1 index of the microbiota; (**E**) PD whole tree index of the microbiota; (**F**) Good’s coverage of the samples; (**G**) Principal coordinate analysis profile of the microbiota; and (**H**) relative abundances of the dominant phyla in the microbiota. *, *p* < 0.05; **, *p* < 0.01; ***, *p* < 0.001.

**Figure 3 microorganisms-09-01770-f003:**
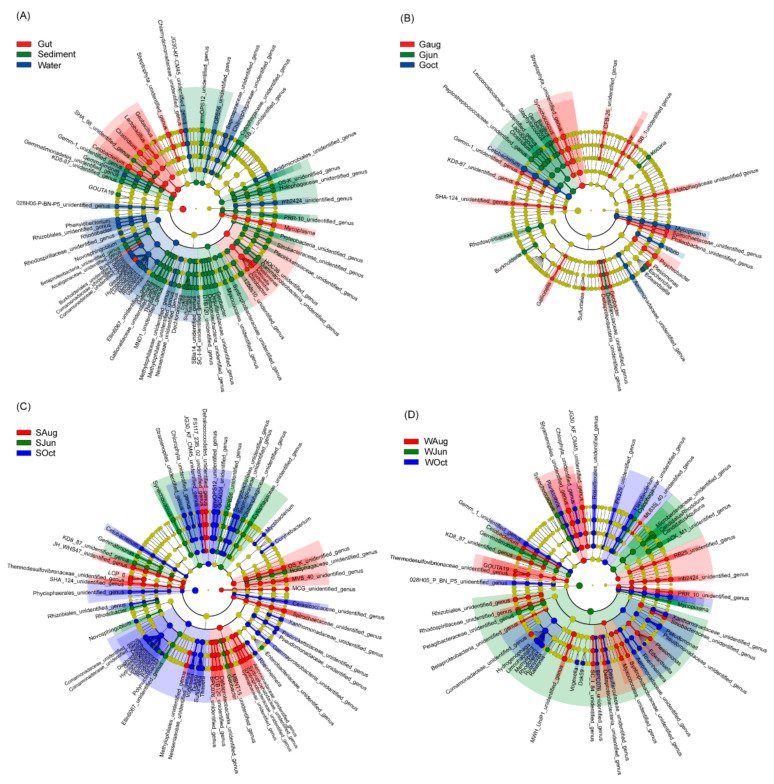
Cladogram plots illustrating significantly different genera (**A**) among different habitats and among different sampling times for microbiota analysis of (**B**) the gut of largemouth bass, (**C**) sediment, and (**D**) pond water. The results with LDA scores (log 10) > 2 were considered statistically significant.

**Figure 4 microorganisms-09-01770-f004:**
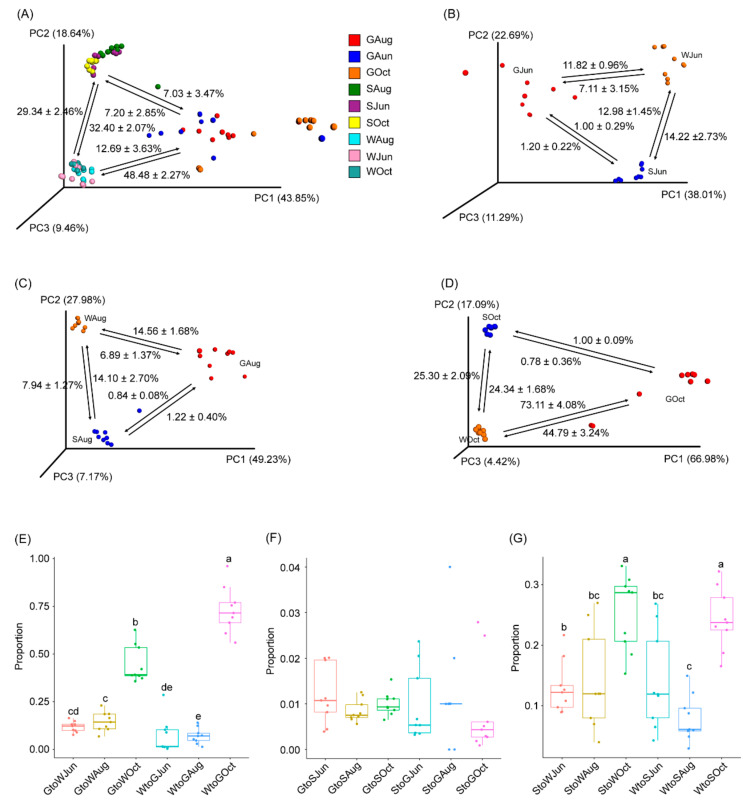
PCoA profiles along with source tracking results highlight the connection between the largemouth bass gut microbiota with pond water and sediment microbiota. (**A**) All samples, (**B**) samples collected in June, (**C**), samples collected in August, (**D**) samples collected in October, (**E**) the proportion of bacteria subjected to exchange between the gut microbiota and pond water microbiota, (**F**) the proportion of bacteria subjected to exchange between the gut microbiota and pond sediment microbiota, and (**G**) the proportion of bacteria subjected to exchange between pond water and sediment microbiota. GJun, GAug, and GOct indicate largemouth bass gut samples collected in June, August, and October, respectively. SJun, SAug, and SOct indicate pond sediment samples collected in June, August, and October, respectively. WJun, WAug, and WOct indicate pond water samples collected in June, August, and October, respectively. Different lowercase letters at the top of the box indicate significant differences between the two groups.

## Data Availability

The merged DNA sequences were deposited in the Genome Sequence Archive with accession number CRA004446 (https://ngdc.cncb.ac.cn/gsa/browse/CRA004446, accessed on 6 July 2021).

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
