# Peer review of "Connection between the Gut Microbiota of Largemouth Bass (Micropterus salmoides) and Microbiota of the Pond Culture Environment"

_microorganisms, 2021, doi:10.3390/microorganisms9081770_

Round 1
Reviewer 1 Report
The proposed work is innovative and interesting for the discipline both from a theoretical and practical point of view. Studying the intestinal bacterial composition of Micropterus salmoides in relation to the environmental one, to favor and understand the interactions between the two in order to optimize the breeding techniques of the species, is an approach that in my opinion is truly valid and relevant.
The introduction is well articulated, it clearly expresses the intentions of the project and the information reported on the percentages of farmed fish significantly support the importance of this project.
The explanation of the materials and methods in sub-paragraph 2.1 is correct and simple, however there is no data on the volume of water actually present in the 3 lakes. Only the surface does not show the real number of animals per cubic meter. the average size (weight and length) of the animals at the end of the breeding cycle must also be entered.
Sub-paragraph 2.2 requires a summary table that highlights the concentrations of all the analytes sought. Why it was not measured Biochemical Oxygen Demand at 5 days?
The results paragraph is well set up, discursive and consistent. The graphs are also well done and reflect the presentation of the results well, also supported by a good statistical analysis.
Discussions: this section is also consistent with the rest of the article, but there is no study proposal throughout the year or the average life of a fish in order to attribute the greatest quantities of nutrients present in the waters to specific periods and specific size classes.
The article as a whole is well structured, it starts from a very valid project idea to increase the production of fish in the lakes used for this purpose. Some parts could be improved as suggested
Author Response
Comments
The proposed work is innovative and interesting for the discipline both from a theoretical and practical point of view. Studying the intestinal bacterial composition of Micropterus salmoides in relation to the environmental one, to favor and understand the interactions between the two in order to optimize the breeding techniques of the species, is an approach that in my opinion is truly valid and relevant.
The introduction is well articulated, it clearly expresses the intentions of the project and the information reported on the percentages of farmed fish significantly support the importance of this project.
Response
Thank you very much for your affirmation of our research work. We will continue to pay attention to and study the structural characteristics and functions of fish and habitat microbiota. We very appreciate your comments on our manuscript. The comments are very helpful for us to improve our manuscript. We have revised our manuscript according to your and another reviewer’s comments. Thank you again.
Comments
The explanation of the materials and methods in sub-paragraph 2.1 is correct and simple, however there is no data on the volume of water actually present in the 3 lakes. Only the surface does not show the real number of animals per cubic meter. the average size (weight and length) of the animals at the end of the breeding cycle must also be entered.
Response
Thank you for your comments. We have added the depth of pond water in our revised manuscript according to your comments. Moreover, we had provided the body lengths and body weights of the largemouth bass in the 3.1 subsection of our manuscript.
Comments
Sub-paragraph 2.2 requires a summary table that highlights the concentrations of all the analytes sought. Why it was not measured Biochemical Oxygen Demand at 5 days?
Response
Thank you very much for your comments. We have added a summary table as Figure S1 in the Results section. Due to the limitation of experimental conditions, we did not determine the BOD.
Comments
The results paragraph is well set up, discursive and consistent. The graphs are also well done and reflect the presentation of the results well, also supported by a good statistical analysis.
Response
Thank you for your comments.
Comments
Discussions: this section is also consistent with the rest of the article, but there is no study proposal throughout the year or the average life of a fish in order to attribute the greatest quantities of nutrients present in the waters to specific periods and specific size classes.
Response
Thank you very much for your comments. Largemouth bass commercially cultured in ponds usually has a breeding cycle of only 5-8 months. It was fed with commercial feed in the whole culture cycle. Therefore, it cannot reflect the greatest quantities of nutrients present in the waters to specific periods and specific size classes.
Comments
The article as a whole is well structured, it starts from a very valid project idea to increase the production of fish in the lakes used for this purpose. Some parts could be improved as suggested.
Response
Thank you very much for your comments. The comments are very helpful for us to improve our manuscript. We have revised our manuscript according to your and another reviewer’s comments.
Reviewer 2 Report
Review of 'Connection between the gut microbiota of largemouth bass (Micropterus salmoides) and microbiota of the pond culture environment ' by Qianfu Liu, Zini Lai, Yuan Gao, Chao Wang, Yanyi Zeng, Erchun Liu, Yongzhan Mai, Wanling Yang and Haiyan Li.
Overall a good study with well thought out laboratory procedures. In this study, the authors studied the gut microbiota in largemouth bass, a cultured species in China, and found significant differences when compared their data with microbial communities in water and sediments. These results are very relevant considering fish aquaculture and its importance for supplying local markets.
Abstract
Pg 1 Ln 21: Suggest changing 'microbiota' to 'microbiota of largemouth bass'
Pg 1 Ln 27: Suggest changing 'interaction' to 'interactions'
Introduction
Pg 2 Ln 52: Suggest changing ' interaction' to 'interactions'
Pg 2 Ln 55: Suggest changing 'this interaction' to 'these interactions'
Pg 2 Ln 58: Suggest changing ' helped' to 'helps"
Materials and Methods
The authors should provide information about the food and feeding regimes of the fish used in the study at each period of sampling.
Results
Pg 3 Ln 142-144: The authors should clarify if these changes reflected growth of the cultured fish or were attributed to other factors.
Pg 4 Ln 158-159: 'the finding was consistent with a previous report [33]'. This fragment should be moved in Discussion.
Discussion
Pg 8 Ln 249: Suggest changing 'interaction' to 'interactions'
Pg 8 Ln 260-261: The authors should provide some hypotheses to explain changes in the gut microbiota of largemouth bass. For example, environmental factors, feeding regimes etc. The authors used different-sized fishes and their metabolism can vary significantly.
Pg 8 Ln 269: Suggest changing 'not present' to 'not being present'
Pg 9 Ln 300: Suggest changing ' constitute to' to 'constitute'
Pg 9 Ln 301: Suggest changing 'a stable' to 'stable'
Pg 9 Ln 304: Suggest changing 'increase' to 'an increase'
Pg 9 Ln 310: Suggest changing 'progression' to 'development'
Author Response
Comments
Overall a good study with well thought out laboratory procedures. In this study, the authors studied the gut microbiota in largemouth bass, a cultured species in China, and found significant differences when compared their data with microbial communities in water and sediments. These results are very relevant considering fish aquaculture and its importance for supplying local markets.
Response
We very appreciate your comments on our manuscript. The comments are very helpful for us to improve our manuscript. We have revised our manuscript according to your and another reviewer’s comments. Thank you again.
Comments
Abstract
Pg 1 Ln 21: Suggest changing 'microbiota' to 'microbiota of largemouth bass'
Response
Thank you very much for your comment. We have revised the word according to your comment.
Comments
Pg 1 Ln 27: Suggest changing 'interaction' to 'interactions'
Response
Thank you very much for your comment. We have revised the word according to your comment.
Comment
Introduction
Pg 2 Ln 52: Suggest changing ' interaction' to 'interactions'
Response
Thank you very much for your comment. We have revised the word according to your comment.
Comment
Pg 2 Ln 55: Suggest changing 'this interaction' to 'these interactions'
Response
Thank you very much for your comment. We have revised the words according to your comment.
Comment
Pg 2 Ln 58: Suggest changing ' helped' to 'helps"
Response
Thank you very much for your comment. We have revised the word according to your comment.
Comment
Materials and Methods
The authors should provide information about the food and feeding regimes of the fish used in the study at each period of sampling.
Response
Thank you very much for your comment. During the experimental period, the fish were fed by commercial puffed compound feed for largemouth bass (Guangdong Haida Group Co., Ltd, China) at 9:00 and 16:00 h at a daily rate of 3% body weight. We have provided the information according to your comment.
Comment
Results
Pg 3 Ln 142-144: The authors should clarify if these changes reflected growth of the cultured fish or were attributed to other factors.
Response
These changes reflected growth of the cultured fish. Thank you for your comment. We have clarified it according to your comment.
Comment
Pg 4 Ln 158-159: 'the finding was consistent with a previous report [33]'. This fragment should be moved in Discussion.
Response
Thank you very much for your comment. Although the fragment is more like a discussion, since it is not necessary to expand to discuss, we think it is more appropriate to leave the fragment in the results section.
Comment
Discussion
Pg 8 Ln 249: Suggest changing 'interaction' to 'interactions'
Response
Thank you very much for your comment. We have revised the word according to your comment.
Comments
Pg 8 Ln 260-261: The authors should provide some hypotheses to explain changes in the gut microbiota of largemouth bass. For example, environmental factors, feeding regimes etc. The authors used different-sized fishes and their metabolism can vary significantly.
Response
Thank you very much for your comment. We have revised our manuscript according to your comment.
Comment
Pg 8 Ln 269: Suggest changing 'not present' to 'not being present'
Response
Thank you very much for your comment. We have revised the word according to your comment.
Comment
Pg 9 Ln 300: Suggest changing ' constitute to' to 'constitute'
Response
Thank you very much for your comment. We have revised the word according to your comment.
Comment
Pg 9 Ln 301: Suggest changing 'a stable' to 'stable'
Response
Thank you very much for your comment. We have revised the word according to your comment.
Comment
Pg 9 Ln 304: Suggest changing 'increase' to 'an increase'
Response
Thank you very much for your comment. We have revised the word according to your comment.
Comment
Pg 9 Ln 310: Suggest changing 'progression' to 'development'
Response
Thank you very much for your comment. We have revised the word according to your comment.